# The Notch-mediated hyperplasia circuitry in *Drosophila* reveals a Src-JNK signaling axis

**Diana M Ho[1]\*, SK Pallavi[1,2], Spyros Artavanis-Tsakonas[1,3]\***

[1]Department of Cell Biology, Harvard Medical School, Boston, United States;
[2]Translational Health Science and Technology Institute, Faridabad, India;
[3]Biogen Idec, Cambridge, United States

**Abstract** Notch signaling controls a wide range of cell fate decisions during development and disease via synergistic interactions with other signaling pathways. Here, through a genome-wide genetic screen in *Drosophila*, we uncover a highly complex Notch-dependent genetic circuitry that profoundly affects proliferation and consequently hyperplasia. We report a novel synergistic relationship between Notch and either of the non-receptor tyrosine kinases Src42A and Src64B to promote hyperplasia and tissue disorganization, which results in cell cycle perturbation, JAK/STAT signal activation, and differential regulation of Notch targets. Significantly, the JNK pathway is responsible for the majority of the phenotypes and transcriptional changes downstream of Notch-Src synergy. We previously reported that Notch-Mef2 also activates JNK, indicating that there are commonalities within the Notch-dependent proliferation circuitry; however, the current data indicate that Notch-Src accesses JNK in a significantly different fashion than Notch-Mef2.

**\*For correspondence:**
diana_ho@hms.harvard.edu
(DMH); artavanis@hms.harvard.
edu (SA-T)

**Competing interests:** The authors declare that no competing interests exist.

**Reviewing editor:**
K VijayRaghavan, National Centre for Biological Sciences, Tata Institute for Fundamental Research, India

## Introduction

A relatively small number of highly conserved cellular signaling pathways are responsible for a broad array of distinct, specific biological processes in metazoan development. It is clear that these pathways must interact in a combinatorial and context-dependent manner to produce the wide and diverse range of downstream events required for development, homeostasis, and disease.

One of these fundamental signaling mechanisms is the Notch pathway, which is conserved amongst all metazoans and controls a wide range of cell fate decisions (*Artavanis-Tsakonas et al., 1999*; *Louvi and Artavanis-Tsakonas, 2006*). Aberrant Notch signaling levels can lead to developmental defects and various pathological conditions including cancer (*Artavanis-Tsakonas and Muskavitch, 2010*; *Ranganathan et al., 2011*; *Louvi and Artavanis-Tsakonas, 2012*). In addition to the well-documented causative role of activating Notch mutations in T-cell acute lymphoblastic leukemia (*Ellisen et al., 1991*; *Weng et al., 2004*), Notch activity has been positively correlated with a number of solid cancers, including those of the breast, prostate, skin, brain, lung, colon, and pancreas (*Koch and Radtke, 2010*; *Ranganathan et al., 2011*). However, Notch can also act as a tumor suppressor in other contexts (*Dotto, 2008*). The specific mechanisms by which Notch contributes to oncogenesis remain largely opaque, although it seems clear that integration with other genes in a context-dependent fashion are key for mediating the action of Notch and its involvement in pro-oncogenic events as proliferation and metastasis.

Ligand binding to the Notch cell surface receptor results in a series of cleavages that release the active C-terminus, which subsequently translocates to the nucleus and modulates the transcription of target genes (*Artavanis-Tsakonas and Muskavitch, 2010*; *Hori et al., 2013*). Given that Notch is highly pleiotropic, it is clear that the developmentally crucial and highly specific downstream

**eLife digest** The cells within animals are organized into tissues and organs that perform particular roles. To develop and maintain these structures, the ability of individual cells to divide and grow is strictly controlled by the activities of many proteins, including one called Notch. This protein is found in all multicellular organisms and allows cells to communicate with each other. Mutations in the gene that encodes Notch can cause cells to divide excessively and lead to cancer and other diseases.

Notch regulates the growth and division of cells by interacting with many other proteins. For example, Mef2 works with Notch to activate a communication system called the JNK pathway. This pathway is involved in controlling cell division, cell death, and cell movement. However, it is thought that Notch may also interact with other proteins that have not yet been identified.

Now, Ho et al. have conducted a genome-wide screen in fruit flies to find proteins that interact with Notch. The experiments used flies that develop abnormally large eyes because they have an over-active Notch protein. Ho et al. identified hundreds of fruit fly genes that could increase or decrease the size of the flies' eyes in the presence of Notch activity. Many of these genes are known to be involved in development, cell division, or in controlling the activity of other genes.

Ho et al. found that two of these genes encode similar proteins called Src42A and Src64B, which are similar to the Src proteins that are involved in many types of human cancers. The experiments show that both proteins interact with Notch to promote uncontrolled cell division and lead to tissues in the flies becoming more disorganized. The JNK pathway is also activated by Notch working with Src42A or Src64B, but in a different manner to how it is activated by Mef2 and Notch, and with different consequences for cells.

This study provides new insights into how genes work together in order to influence cell division and other events in development. Also, it suggests that Notch activity may regulate the growth of cancers linked with defects in the Src proteins.

responses to Notch signal activation depend on its cellular context and the integration of the signal with other signaling pathways (*Hurlbut et al., 2007*; *Bray and Bernard, 2010*). Indeed, a number of genetic screens in our lab and others have revealed a staggering number of genes that interact with Notch to modulate downstream phenotypes; given that the results from these studies, which encompass various biological processes, have surprisingly little overlap, it seems likely that the full gamut of genes that can genetically interact with and influence Notch has not yet been identified (*Kankel et al., 2007*; *Hurlbut et al., 2009*; *Shalaby et al., 2009*; *Saj et al., 2010*; *Guruharsha et al., 2012*).

We previously reported a synergistic interaction between activated Notch and the transcription factor Mef2, and showed that it activates the JNK signaling pathway (*Pallavi et al., 2012*). In addition to its classical role as a cell stress mediator, JNK, like Notch, plays roles in multiple morphogenetic processes including proliferation, cell death, and cell shape changes (*Rios-Barrera and Riesgo-Escovar, 2013*).

In this work, we report the results of a systematic, genome-wide modifier screen in *Drosophila* to dissect and define the genetic circuitry that interacts with Notch to affect proliferation events. We investigate the mechanism of a novel synergistic interaction via JNK between Notch and the *Drosophila* Src genes, which regulate proliferation, apoptosis, adhesion, and motility and whose human orthologs are abnormally activated in numerous types of primary and metastatic tumors (*Stewart et al., 2003*; *Pedraza et al., 2004*; *Vidal et al., 2007*; *Kim et al., 2009*; *Wheeler et al., 2009*; *Guarino, 2010*).

## Results

### An unbiased genetic screen reveals that more than 300 genes potentially interact with Notch to influence proliferation

We performed a genome-wide screen for modifiers of an activated Notch (N$^{act}$)-induced large eye phenotype to uncover novel genes that interact with Notch to affect proliferation using the Exelixis

collection of insertional mutations, which covers approximately 50% of the genome (*Thibault et al., 2004*; *Kankel et al., 2007*; *Pallavi et al., 2012*). We screened for enhancement or suppression of the large eye phenotype (*Figure 1A*). As a result, we identified 360 *Drosophila* genes that are predicted to affect Notch-induced proliferation in the eye; of particular interest are the 206 genes that have clear human orthologs (*Supplementary file 1*). Gene Ontology (GO) analysis reveals that 42 GO categories are significantly enriched among the 360 genes (*Figure 1B*, and *Supplementary file 2*). The majority of these enriched GO terms fall into three broad categories: genes involved in morphogenesis and development, genes involved in cell division and the cell cycle, and genes involved in transcription. Notably, 84 of the 360 genes did not have any associated GO or INTERPRO annotation. The majority of the genes identified in this screen have not previously been linked to Notch. For example, analysis of known and predicted interactions using the GeneMania platform between Notch and the 31 annotated cell cycle genes shows that only one (*inscuteable, insc*) was previously directly linked to Notch (*Figure 1C*).

This unbiased genetic screen reveals the unexpected complexity of the genetic circuitry capable of influencing proliferation events in combination with Notch signals.

## Src overexpression alleles synergize with activated Notch

We previously reported that the transcription factor Mef2, a gene identified in our screen (*Figure 1A*), synergizes with Notch to induce hyperproliferative and metastatic effects through activation of the JNK signaling pathway (*Pallavi et al., 2012*). We therefore asked whether any of the other genes identified in the screen might also be a component of the Notch/Mef2/JNK signaling axis. We retested 26 of the hits from the screen for JNK activation using qPCR to explore changes in expression of *puckered (puc)*, a direct JNK target (*Martin-Blanco et al., 1998*), and *MMP1*, an indirect target (*Uhlirova and Bohmann, 2006*), in *Drosophila* wing discs in an *MS1096Gal4; UAS-N$^{act}$* background. We found that only two of the 26 lines were able to induce both *puc* and *MMP1* (*Supplementary file 1*). These two lines were *d08184* (predicted to overexpress *Eip75EF*) and *d10338* (predicted to overexpress *Src42A*). Of these two lines, *d10338* was a much stronger activator of both *puc* and *MMP1*.

The combination of *d10338* with *UAS-N$^{act}$*, under the *E1Gal4* driver, results in a strong enhancement (*Figure 2A*) of the N$^{act}$ large eye phenotype (*Figure 2C*). We also often observe outgrowths of eye tissue protruding from the borders of the eye (arrow in *Figure 2A*). Notably, *d10338* alone produced smaller eyes (*Figure 2B*) than wild-type controls (*Figure 2D*).

We corroborated that gain-of-function of Src42A is indeed responsible for the synergy by repeating the eye experiment with a constitutively active *Src42A* allele (*UAS-Src42A$^{CA}$*) (*Tateno et al., 2000*). Indeed, Src42A$^{CA}$ also causes hyperplastic eye growth in combination with N$^{act}$ (*Figure 2E*) and reduced eye size when expressed on its own (*Figure 2F*).

As expected, Gal4-driven expression of *d10338* results in an upregulation of both the Src42A gene product and the active, phosphorylated form of Src in *vgGal4/d10338* wing discs (*Figure 2—figure supplement 1*).

*VgGal4*-driven expression of *UAS-N$^{act}$* and *UAS-Src42A$^{CA}$* in the wing disc also resulted in a hyperplastic phenotype; the wing discs are not only overgrown but also noticeably disorganized, with a characteristic 'crumpled ball' phenotype (*Figure 2I*). Furthermore, these larvae fail to pupate and develop a 'giant larvae' phenotype. Src42A$^{CA}$ alone causes disorganization but no increase in overall disc size (*Figure 2J*), whereas N$^{act}$ alone causes increased disc size but minimal apparent disorganization (*Figure 2K*).

There are two distinct Src family members in *Drosophila*, *Src42A* and *Src64B*. Previous reports have shown that they are both widely expressed and may be redundant in many cases (*Takahashi et al., 1996*; *Tateno et al., 2000*; *Takahashi et al., 2005*). Therefore, we asked whether Src64B also synergizes with N$^{act}$. Consistent with the notion that Src42A and Src64B function similarly, we find that Src64B also interacts with N$^{act}$ to produce hyperplastic eyes with outgrowths and overgrown, disorganized wing discs (*Figure 2G* and *Figure 2—figure supplement 2*).

## Involvement of JNK signaling

In order to confirm that JNK signaling acts downstream of Notch and Src (henceforth N/Src), we used the *puc-LacZ* reporter to visualize JNK signal activation in vivo (*Martin-Blanco et al., 1998*).

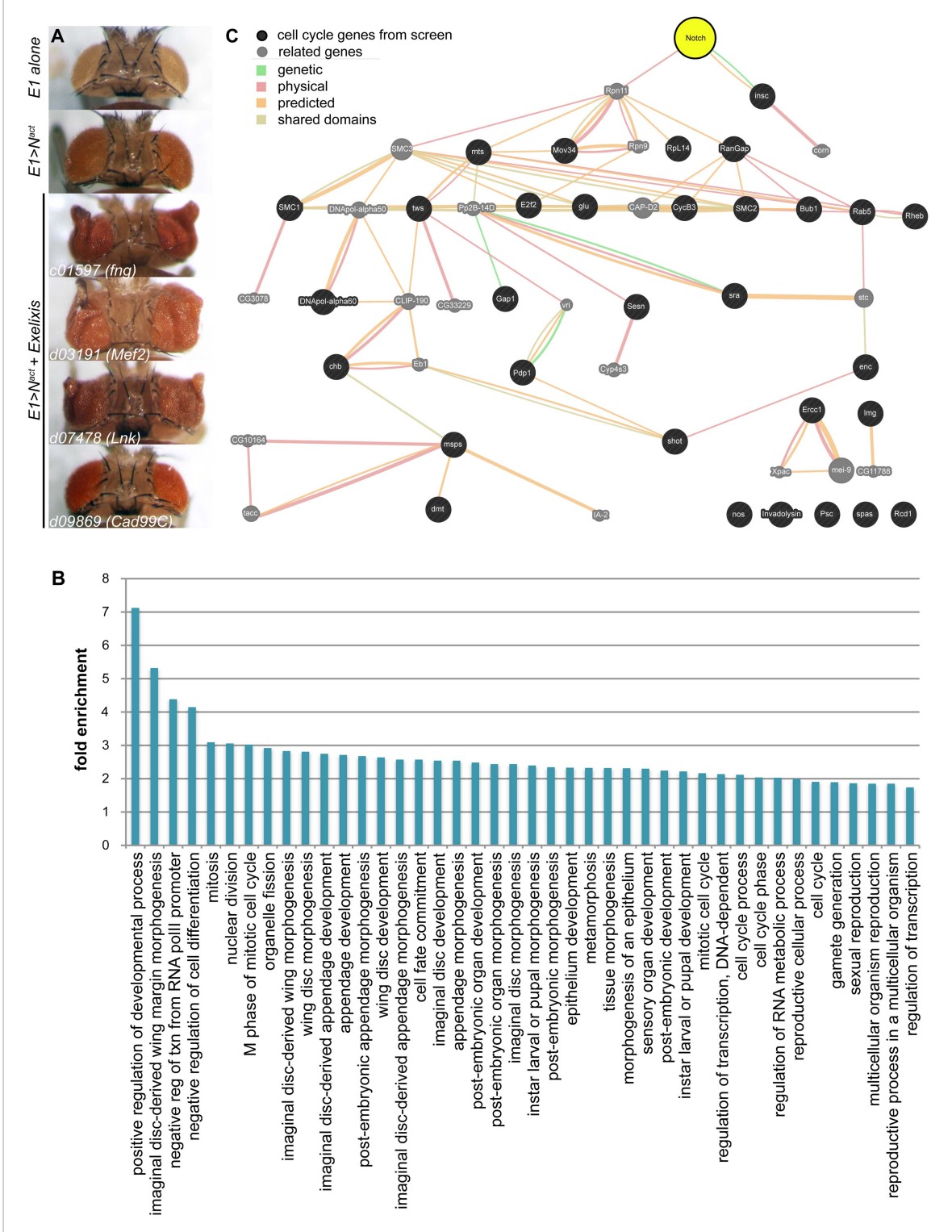

**Figure 1**. A genetic screen for modifiers of Notch-induced proliferation in the *Drosophila* eye. (**A**) Examples of screen phenotypes. *E1>N^act^* results in larger eyes (second panel), compared to wild-type (*E1Gal4* alone) controls (top panel). Examples of three enhancers, *c01597* (fng), *c03191* (Mef2), and *d07478* (Lck), and one suppressor, *d09869* (Cad99C), are shown. (**B**) Analysis of enrichment of GO terms among the 360 *Drosophila* genes identified in the screen. Only enriched terms with corrected p-value < 0.05 (using Benjamini–Hochberg correction) are shown. For numerical p-values, please see *Supplementary file 2*. (**C**) Gene association analysis among cell cycle genes identified in the genetic screen. Genetic interactions, physical interactions, predicted interactions, and shared protein domains were mapped using GeneMania (www.genemania.org) between the 31 cell cycle genes from our screen (black circles) and Notch (yellow). Genes labeled with grey circles are part of the network but were not identified in our screen.

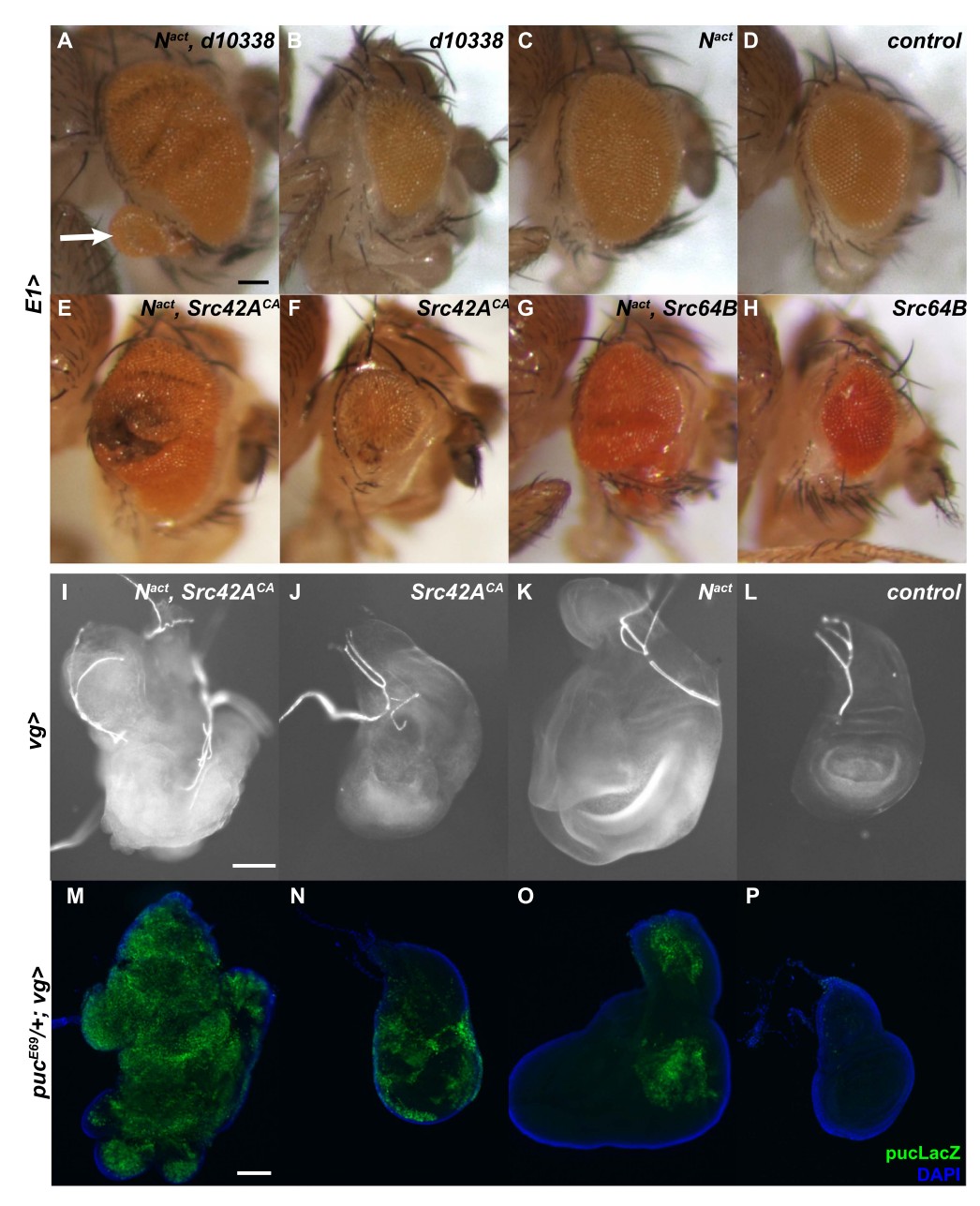

**Figure 2**. Synergy between Notch and Src in the eye and wing causes hyperplastic phenotypes and activates JNK. (A–H) Various *UAS-Src* constructs were driven by *E1Gal4* along with UAS-N^act in the developing eye. When *d10338*, an Exelixis allele that causes Gal4-dependent overexpression of Src42A, and N^act are coexpressed (A), the N^act large eye phenotype (C) is enhanced; in addition, occasional outgrowths of eye tissue can be seen (arrow). Note that d10338 alone (B) results in decreased eye size, whereas N^act alone (C) results in increased eye size compared to the control (D). Src42A^CA and Src64B both cause a similar phenotype (E, G) when coexpressed with N^act under *E1Gal4*, and both also result in decreased eye size in the absence of N^act (F, H). (I–L) *UAS-N*^act and *UAS-Src42A*^CA were driven in the developing wing using the *vgGal4* driver. When N^act and Src42A^CA are co-expressed (I), wing discs are overgrown compared to either Src42A^CA (J) or N^act (K) alone and display a characteristic 'crumpled ball' phenotype indicative of tissue disorganization and cell migration. Note that Src42A^CA alone (J) causes disorganization but not overgrowth. (M–P) *Puc-LacZ* reporter assay for JNK signal activation in wing discs expressing UAS constructs as indicated under the *vgGal4* driver in a *puc*^E69/+ background. Coexpression of N^act and Src42A^CA (M) causes strong, global activation of the *pucLacZ* reporter. In contrast, expression of either gene alone (N, O) causes weaker activation that is limited in scope. Scale bars: 100 μm.

*Figure 2. continued on next page*

*Figure 2. Continued*

The following figure supplements are available for figure 2:

**Figure supplement 1**. *d10338* is a UAS allele of *Src42A*.

**Figure supplement 2**. Src64B also synergizes with N[act] in the wing disc.

Coexpression of N[act] and Src42A[CA] in *vgGal4* wing discs results in strong, widespread LacZ expression (*Figure 2M*). In contrast, N[act] or Src42A[CA] alone each induced far weaker, spatially restricted *puc-LacZ* activation (*Figure 2N,O*).

Given that previous studies associated increased JNK signaling with both invasiveness and apoptosis (*Uhlirova and Bohmann, 2006*; *Pallavi et al., 2012*), we tested for expression of MMP1, a matrix metalloprotease associated with invasive phenotypes and cleaved caspase 3 (cl-casp3), an apoptotic marker. Coexpression of N[act] and Src42A[CA] caused high levels of both MMP1 and cl-casp3 (*Figure 3A,G*). This is in striking contrast to N[act]+Mef2, which results in robust MMP1 activation but little to no apoptosis, consistent with our previous report (*Figure 3F,L*) (*Pallavi et al., 2012*).

To examine whether JNK signaling is responsible for the observed phenotypes, we used a dominant negative form of the *Drosophila* JNK gene *Basket* (*UAS-Bsk[DN]*) to block JNK signaling in N/Src wing discs. We found that *Bsk[DN]* reduced both MMP1 and cl-casp3 to near-wildtype levels (*Figure 3E,K*). In order to rule out the formal possibility that the observed rescue is caused by titration of Gal4 by an additional UAS rather than by a *bona fide* effect of the Bsk[DN] transgene, we coexpressed *UAS-N[act]* and *UAS-Src42A[CA]* with *UAS-GFP* and observed no discernible rescue (*Figure 3—figure supplement 1*).

In our earlier work, we reported that N/Mef2 activates JNK through the TNF ligand *eiger (egr)* (*Pallavi et al., 2012*); we therefore asked whether this is also the case for N/Src. We find that N[act]+Src42A[CA]

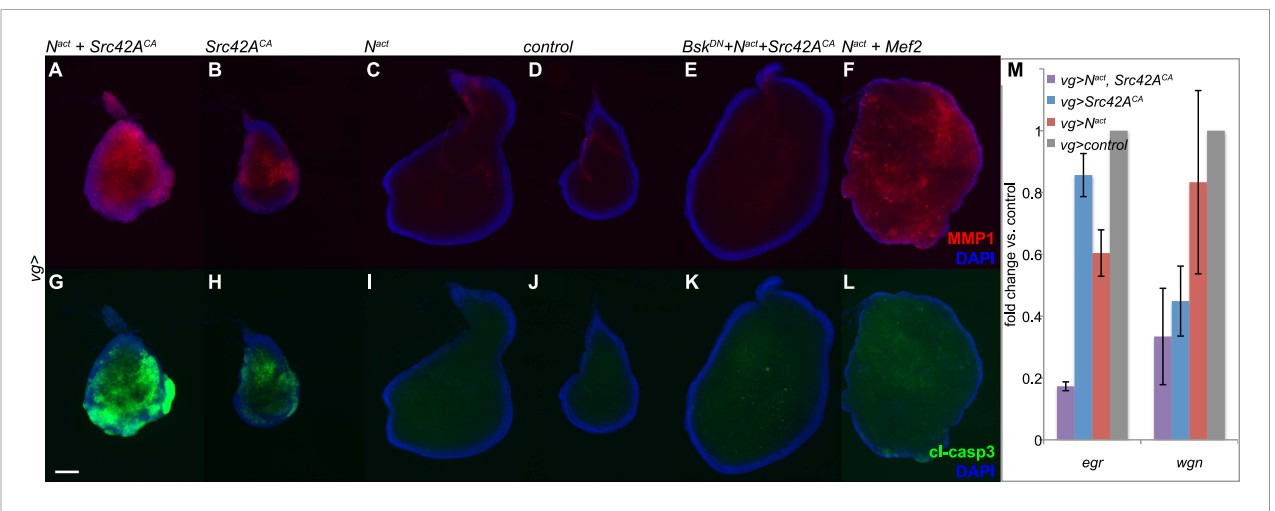

**Figure 3**. N/Src synergy induces both MMP1 and apoptosis. (**A–L**) Immunofluorescence for MMP1 (**A–F**) and cleaved caspase 3 (cl-casp3, **G–L**) in wing discs expressing UAS constructs under *vgGal4*. Together, N[act] and Src42A[CA] cause robust activation of both MMP1 (**A**) and cl-casp3 (**G**), which is strongly reduced by Bsk[DN] (**E, K**). The combination of N[act] and Mef2 results in an increase in MMP1 (**F**) but little effect on cc3 (**L**). (**M**) qPCR for *egr* and *wgn* in wing discs overexpressing genes as indicated under the *vgGal4* driver reveals that both transcripts are strongly downregulated when N[act] and Src42A[CA] are coexpressed. Scale bar: 100 µm.

The following figure supplements are available for figure 3:

**Figure supplement 1**. Gal4/UAS titration does not affect the N/Src phenotype.

**Figure supplement 2**. A heterozygous null mutation of Notch can rescue lethality and phenotype of Src alone.

actually causes a synergistic downregulation of both *egr* and its receptor *wengen (wgn)*, suggesting that unlike the N/Mef2 synergy, N/Src activates JNK via a TNF-independent mechanism (*Figure 3M*).

Since the *vestigial (vg)* gene is known to be a target of Notch (*Klein and Arias, 1998*), we wished to rule out the possibility that the phenotypes we observe could be complicated by an effect of Nact directly on the *vgGal4* driver. Therefore, we repeated the above experiment using the *dppGal4* driver, which is expressed in the anterior-posterior boundary of the wing disc. Just as with *vgGal4*, *dppGal4*-driven Nact+Src42ACA induces MMP1 and cl-casp3 in the wing disc, and this effect is rescuable by BskDN (*Figure 4*). Furthermore, we included a UAS-GFP transgene in these experiments to mark the domain of transgene expression and determine whether or not the effects we see are cell-autonomous. Indeed, we find that some GFP-positive cells, notably those in the far ventral region of the disc, do not obviously express either MMP1 or cl-casp3 (green arrows in *Figure 4*). Additionally,

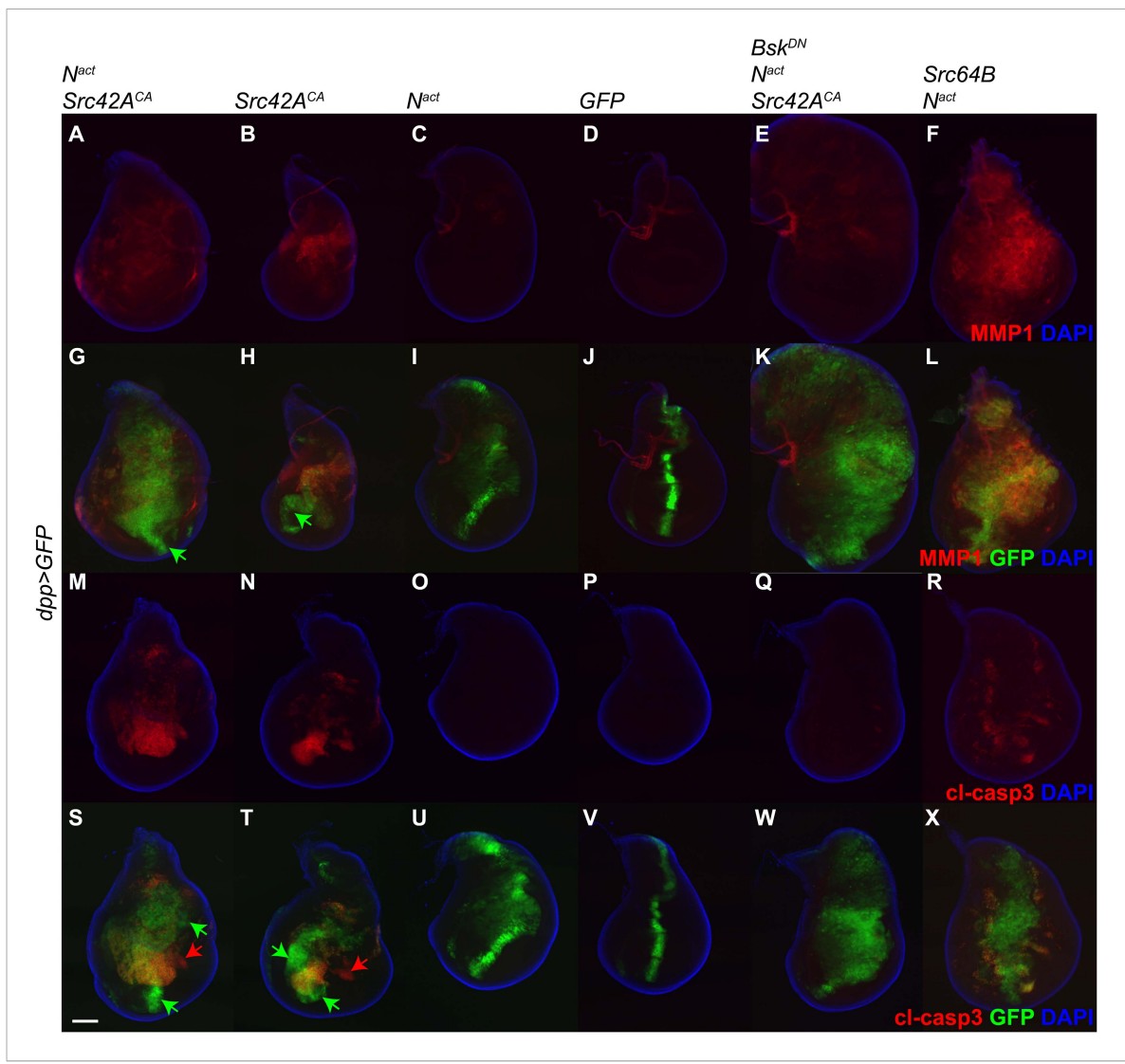

**Figure 4**. *dpp-Gal4* driven expression of Nact and Src42ACA also upregulates MMP1 and induces apoptosis. UAS transgenes as indicated were driven with *dppGal4* along with *UAS-GFP* at 18°C. Controls express an extra copy of *UAS-GFP*. Wing discs were stained with anti-MMP1 (**A–L**) or anti-cleaved caspase 3 (cl-casp3, **M–X**). The combination of Nact and Src42ACA induces both MMP1 (**A**, **G**) and cl-casp3 (**M**, **S**), and Src42ACA alone does the same to a lesser extent (**B**, **H**, **N**, **T**). Green arrows: GFP positive cells that do not express MMP1 (**G**, **H**) or cl-casp3 (**S**, **T**). Red arrows: cl-casp3-positive cells that do not express GFP, indicating a potentially non-cell-autonomous effect. This effect can be largely rescued with BskDN (**E**, **K**, **Q**, **W**). Similarly, the combination of Src64B and Nact also induces both MMP1 (**F**, **L**) and cl-casp3 (**R**, **X**). Scale bar: 100 μM.

some apoptosis is detected in GFP-negative domains of both N$^{act}$+Src42A$^{CA}$ and Src42A$^{CA}$ wing discs (red arrows in *Figure 4*), suggesting a non-cell-autonomous effect, although we cannot rule out the possibility that since these cells are dying, they have also stopped expressing GFP.

These observations were also corroborated by determining that Src64B, in concert with N$^{act}$, also strongly activates both MMP1 and cl-casp3 (*Figure 4F,L,R,X*). Interestingly, it appears to induce correspondingly higher levels of MMP1 and lower levels of apoptosis than Src42A$^{CA}$; we note, however, that the Src64B allele we used is a WT allele, whereas the Src42A$^{CA}$ is constitutively active.

Taken together, these findings reveal that Notch and Src act together to induce JNK signaling and subsequent downstream consequences such as MMP1 activation and induction of apoptosis, but that this JNK activation differs subtly from that induced by Notch and Mef2.

It is noteworthy that Src42A or Src64B alone can often generate apparently weaker versions of the N/Src synergistic phenotype (*Figures 2–4*). Notch is endogenously active in both the eye and wing discs during the time when the Src transgenes are expressed in our experiments (*Johnston and Edgar, 1998*; *Baonza and Garcia-Bellido, 2000*; *Baonza and Freeman, 2005*; *Herranz et al., 2008*). We therefore suspected that the observed effects of Src alone could be caused by synergy between exogenous Src and endogenous Notch. To test this, we used the N$^{55e11}$ heterozygous mutation to decrease levels of endogenous Notch in the developing wing. When *UAS-Src64B* is driven by *vgGal4* in a wild-type background (*FM7C/+;vg-Gal4/UAS-Src64B*), we observe a high degree of lethality, with the few escapers (n = 16 over four independent experiments) displaying small, shriveled, vestigial wings. In contrast, N$^{55e11}$/+;*vgGal4/UAS-Src64B* siblings demonstrated reduced lethality (n = 126) and fully extended, notched wings similar to the notched wing phenotype of N$^{55e11}$/+;*vgGal4/+* controls (*Figure 3—figure supplement 2A–D*). MMP1 and cl-casp3 induced by Src64B were also significantly reduced in N$^{55e11}$/+;*vgGal4/UAS-Src64B* wing discs (*Figure 3—figure supplement 2E–H*).

A similar rescue occurs when we combine the *d10338* Src42A allele with *UAS-Notch$^{RNAi}$* (N$^{RNAi}$). *d10338* alone, when driven with *vgGal4*, is largely lethal, with the few escapers (n = 3, compared to 62 balancer siblings) having only wing stumps. In contrast, when *UAS-N$^{RNAi}$* is added, lethality is largely rescued (n = 67, vs 108 balancer siblings) and the wings form long, thin spikes similar to those generated by N$^{RNAi}$ alone (*Figure 3—figure supplement 2I–K*).

We finally note that the above experiment does not rule out the possibility that the observed rescue could be caused by an effect of N$^{55e11}$/+ or *UAS-N$^{RNAi}$* on the *vgGal4* driver itself. Unfortunately, attempts to repeat these experiments using other Gal4 drivers (*C96Gal4* or *dppGal4*) or Src-activating mutants (*Csk$^{j1d8/j1d8}$* or *Csk$^{j1d8/+}$*, *puc$^{E69/+}$* [*Langton et al., 2007*]) were unsuccessful due to high levels of lethality.

## N/Src synergy perturbs the cell cycle

The cell cycle is often misregulated during hyperplastic or cancerous growth. We thus performed a DNA content analysis to examine the cell cycle distribution of GFP-positive cells from *vgGal4;UAS-GFP* wing discs expressing Src64B and N$^{act}$. Whereas wild type and N$^{act}$ cells have 23% and 25% of cells in G1 phase respectively, N$^{act}$+Src64B cells showed a complete loss of G1 phase. Src64B alone caused a partial loss of G1 phase (4%). In both cases, a corresponding increase in the proportion of S phase cells was also observed. The effect on the cell cycle appears to be dependent on JNK signaling, as blocking JNK with Bsk$^{DN}$ strongly reversed the G1 bypass (19% of cells in G1). Both Src42A alleles (Src42A$^{CA}$ and d10338) produced cell cycle distribution profiles (2% and 3% in G1 phase respectively) similar to Src64B when coexpressed with N$^{act}$. Likewise, wing disc cells coexpressing N$^{act}$ and Mef2 displayed a decreased number of cells in G1 phase (5%) (*Figure 5A*). Cells from *dppGal4;UAS-GFP* wing discs, similar to those from *vgGal4; UAS-GFP* discs, coexpressing Src64B and N$^{act}$ also displayed a similar, JNK-dependent loss of the G1 peak (*Figure 5—figure supplement 1*).

Cell cycle arrest in G1 phase is a hallmark of cells in the Zone of Non-proliferating Cells (ZNC) located in the D-V boundary of the wing disc (*Johnston and Edgar, 1998*). If N/Src could cause G1 bypass, then, we reasoned, it might also cause ZNC disruption. We visualized the ZNC by incorporating EdU, which labels cells in S phase, in *dppGal4;UAS-GFP* wing discs. We used the *d10338* Src42A allele, as the more modest disruption of disc organization caused by this weaker allele allowed us to better identify the ZNC. Control ZNC cells that are arrested in G1 do not enter S phase and therefore do not incorporate EdU (*Figure 5E*); N$^{act}$ alone causes ZNC expansion and a partially non-cell autonomous increase in EdU incorporation in the dorsal region of the disc (*Figure 5D*), as

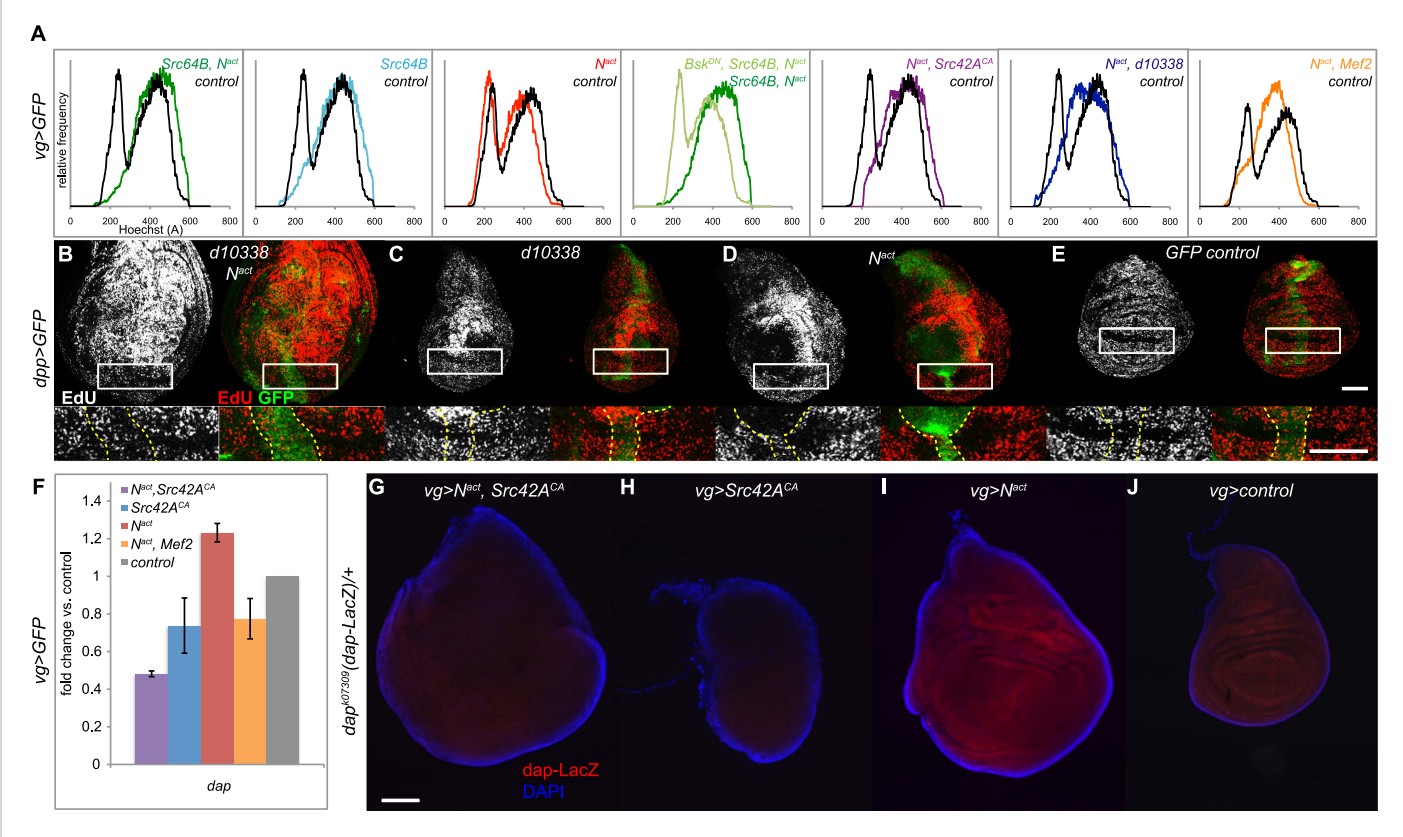

**Figure 5**. N/Src synergy disrupts the cell cycle. (**A**) DNA content analysis was performed on Hoechst-labeled dissociated cells from *vgGal4;UAS-GFP* wing discs expressing *UAS-Src64B;UAS-N^act* (dark green trace), *UAS-Src64B* (light blue), *UAS-N^act* (red), WT control (black), *UAS-Bsk^DN;UAS-Src64B;UAS-N^act* (light green), *UAS-N^act;UAS-Src42A^CA* (purple), *UAS-N^act;d10338* (dark blue) or *UAS- N^act;UAS-Mef2* (orange). Comparative histograms show relative frequencies on the y-axis, normalized to total number of counts for each sample. (**B–E**) EdU incorporation assay in *dppGal4;UAS-GFP* wing discs expressing *d10338;UAS-N^act* (**B**), *d10338* (**C**), *UAS-N^act* (**D**), or *UAS-GFP* (**E**) at 22°C. A closeup of the areas denoted by boxes is shown below each image, and the GFP-positive area is marked with dotted yellow lines. Whereas *UAS-N^act* alone expands the ZNC (zone of non-proliferating cells) and also non-cell-autonomously induces proliferation in the dorsal-posterior region of the disc, thus increasing the size of the dorsal compartment (**D**), the combination of *d10338* and *UAS-N^act* eliminates the expansion of the non-proliferative zone and causes cells within the ZNC proper to begin incorporating EdU; furthermore, the area of increased proliferation in the dorsal compartment appears to be expanded (**B**). (**F–J**) N^act and Src42A^CA together cause a reduction in *dacapo (dap)* levels. (**F**) qPCR for *dap* expression in wing discs expressing N^act and/or Src42A^CA or Mef2 under the *vgGal4* driver. (**G–J**) A *dap-LacZ* reporter assay was used to visualize *dap* expression in *vgGal4* wing discs in a *dap^k07309*/+ background. Both N^act and Src42A^CA together (**G**) and Src42A^CA alone (**H**) show a reduction in dap-LacZ compared to both N^act alone (**I**) and *vgGal4* controls (**J**). Scale bars: 100 μM.

The following figure supplement is available for figure 5:

**Figure supplement 1**. Elimination of G1 phase of the cell cycle also occurs in *dppGal4* wing discs expressing N^act and Src64B.

previously reported (*Go et al., 1998*; *Johnston and Edgar, 1998*; *Herranz et al., 2008*). In contrast, discs expressing N^act and d10338 show a reduced ZNC with correspondingly more EdU-labeled cells in the D–V boundary, and the area of Notch-induced increased proliferation in the dorsal region is expanded, extending all the way down to the ZNC and greatly increasing the size of the dorsal compartment (*Figure 5B*).

Given that the cyclin-dependent kinase (CDK) inhibitor *dacapo (dap)* blocks G1 to S transition and is important for cell cycle exit in G1 (*de Nooij et al., 1996*; *Lane et al., 1996*) we asked whether N/Src affects transcription of *dap*. qPCR reveals that N^act+Src42A^CA expression is indeed associated with a downregulation of both Notch-induced and endogenous *dap* transcription; we note that N^act+Mef2 similarly reduced *dap* levels, although to a lesser extent (*Figure 5F*). We corroborated the qPCR result using the *dap^k07309* enhancer trap line, which functions as a *dap-LacZ* reporter (*Mitchell et al., 2010*). N^act alone activates *dap-LacZ*, and N^act+Src42A^CA suppresses not only this increase but also the

endogenous expression of the reporter (*Figure 5G,I,J*). Src42A^CA alone also causes a significant decrease in reporter expression (*Figure 5H*).

We conclude that N/Src synergy results in bypassing the G1 phase of the cell cycle, likely via the downregulation of the CDK inhibitor *dacapo*.

## JNK and JAK/STAT signaling pathways are downstream of N/Src synergy

Disorganized, hyperplastic growth has been associated with JAK/STAT signaling in *Drosophila*; furthermore, both Src and Notch individually can activate JAK/STAT (*Tateno et al., 2000*; *Read et al., 2004*; *Reynolds-Kenneally and Mlodzik, 2005*). We thus probed whether N/Src expression could affect JAK/STAT signaling.

To assess JAK/STAT signal activation, we used qPCR to measure expression levels of the JAK/STAT ligands *unpaired/outstretched (upd/os), unpaired2 (upd2), and unpaired 3 (upd3)*. We found that all three *upd* genes were strongly upregulated in a synergistic manner by N^act+Src42A^CA; furthermore, this upregulation was largely suppressed by the addition of Bsk^DN, indicating that it is dependent on JNK signals. N^act+Mef2 also activated the *upd* ligands, but to a far lesser extent (*Figure 6A*). An *upd-LacZ* reporter line reveals similar results, with the combination of *UAS-N^act* and *UAS-Src42A^CA* driven by *vgGal4* inducing strong reporter activation compared to either gene alone or controls expressing *vgGal4* alone (*Figure 6B–E*).

To directly visualize JAK/STAT signal activation in vivo, we used the *10XSTATGFP* reporter (*Bach et al., 2007*). We find that, whereas Src42A^CA or N^act alone each caused weak *10XSTATGFP* activation, the combination of the two resulted in strong reporter activation throughout the disc (*Figure 6G–J*). Notably, the patterns of expression of *10XSTATGFP* and *upd-LacZ* are similar. Moreover, consistent

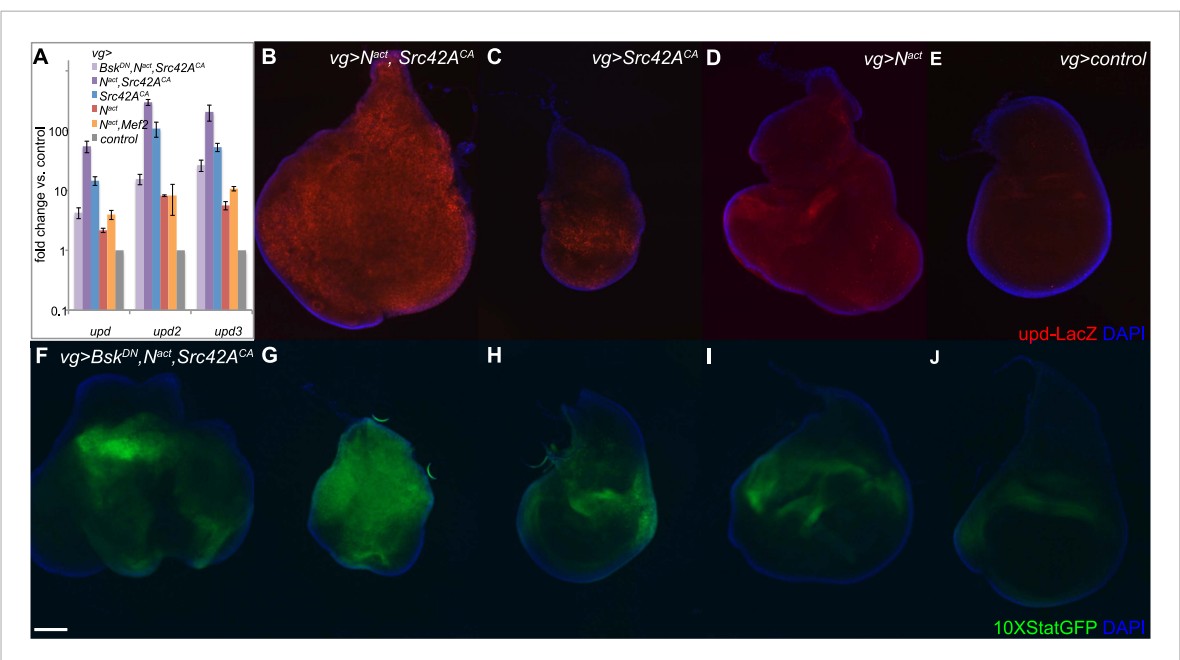

**Figure 6**. N/Src synergy activates the JAK/STAT signaling pathway. (**A**) qPCR for *unpaired* family ligands in *vgGal4* discs expressing UAS constructs as indicated. All three *upd* family genes are highly upregulated by the combination of N^act and Src42A^CA (dark purple bars), and this upregulation is dependent upon JNK signaling as Bsk^DN rescues it (lavender bars). Coexpression of N^act and Mef2 (orange bars) induces a much lower level of the *upd* ligands. Note that the y-axis is on a logarithmic scale. (**B–E**) An *upd-LacZ* reporter assay in *vgGal4* wing discs validates the qPCR data and demonstrates that N^act+Src42A^CA causes strong, widespread activation of *upd* transcription (**B**); in contrast, either gene alone (**C**, **D**) causes lower, more restricted levels of *upd* upregulation. (**F–J**) The *10XStatGFP* reporter was used to assess JAK/STAT signal activation in *vgGal4* discs grown at 18°C. N^act+Src42A^CA strongly upregulates *10XStatGFP* (**G**), whereas either gene alone (**H**, **I**) only weakly upregulates the reporter. The addition of Bsk^DN (**F**) reduces the *10XStatGFP* induced by N^act+Src42A^CA (**G**) to levels similar to those of N^act alone (**I**). Note that since the *upd-LacZ* discs were grown at 25°C and the *10XSTATGFP* discs were grown at 18°C, the latter displays a somewhat weaker phenotype, hence the difference in disc size between **B/D** and **G/I**. Scale bar: 100 μm.

with the observation that blocking JNK reduces the expression of the *upd* genes, Bsk$^{DN}$ also reduced *10XSTATGFP* reporter activation (*Figure 6F*).

Thus, N/Src synergy results in activation of JAK/STAT signaling by the upregulation of the *upd* ligands in a JNK-dependent manner.

## Identification of downstream targets of N/Src synergy

To examine the gamut of transcriptional targets downstream of N/Src coexpression, we performed an RNA-sequencing (RNA-seq) analysis in *vgGal4* wing discs. We defined 'synergistic targets' as those genes that were significantly up- or down-regulated (adjusted p-value < 0.05) in N$^{act}$+Src42A$^{CA}$ discs as compared to N$^{act}$ alone, Src42A$^{CA}$ alone, or WT controls. By these criteria, we identified 187 genes, of which 87 were downregulated and 100 were upregulated; the effects on expression of 130 of these genes was reversed by Bsk$^{DN}$ (*Supplementary file 3*), consistent with the notion that their expression is dependent on JNK. We validated 44 of these genes with qPCR; 4/44 were thus determined to be false positives. We note that of these 187 *Drosophila* genes, 120 have clear human orthologs (*Supplementary file 3*).

It is noteworthy that this analysis did not uncover several known N/Src targets (the *upd* genes, *MMP1*, *puc,* and *dap*). We attribute this to the observation that Src alone often causes a milder version of the N/Src phenotype; our analysis is not always sensitive enough to assign significance to relatively small N/Src vs Src differences. For *MMP1, os/upd, upd2,* and *upd3,* the raw data reveals that the N/Src vs Src comparison was small enough to be insignificant, although *upd2* and *upd3* appeared significant prior to false positive correction. For *dap*, we observe a reduction of Notch-induced *dap* with the addition of Src42A$^{CA}$; however, we did not detect the same reduction in endogenous *dap* between N/Src and WT discs that we saw with qPCR and the *dap-LacZ* reporter. We cannot at present explain this discrepancy. Finally, *puc* did not score as a significant target in our analysis, possibly due to a false positive reading in the RNA-seq data for the WT condition, as there is no evidence to suggest that *puc* is highly upregulated in WT discs (*Supplementary file 4*).

One synergistically upregulated gene was *Enhancer of split mγ (E(spl)mγ* or *HLHmgamma*), a member of the *E(spl)* complex and a target of Notch itself. We therefore asked whether other *E(spl)* complex members were similarly affected. Six of the seven members (excepting *E(spl)m5)* are expressed in the wing disc. However, only *E(spl)mγ* was synergistically upregulated. The other five expressed genes in the locus (*E(spl)m8, E(spl)m3, E(spl)m7, E(spl)mβ*, and *E(spl)mδ*) were, as expected, upregulated by N$^{act}$ alone, but, interestingly, this upregulation actually appeared to be suppressed by the addition of Src, sometimes all the way back to WT levels; Bsk$^{DN}$ reversed both the N/Src-induced enhancement of *E(spl)mγ* (albeit weakly) and the suppression of the other five *E(spl)* genes (*Figure 7A*). In contrast, N$^{act}$+Mef2 caused suppression of all of the *E(spl)* genes, including *E(spl)mγ* (*Figure 7A*). We corroborated these observations by using an *E(spl)mγ* reporter consisting of a 234-bp mγ enhancer region, which has been shown to recapitulate the endogenous *E(spl)mγ* expression pattern in the wing disc, fused to LacZ (*Nellesen et al., 1999*). While, as expected, N$^{act}$ causes an increase in the number of cells that express *E(spl)mγ-LacZ*, N$^{act}$+Src42A$^{CA}$ does not; interestingly Src42A$^{CA}$ alone eliminates the endogenous pattern of *E(spl)mγ-LacZ*, possibly due to cell death (*Figure 7—figure supplement 1*). This observation raises the possibility that E(spl)mγ activation by N/Src is driven by a genomic region distinct from the 234-bp mγ enhancer region.

Because the *E(spl)* complex is an important Notch target, we asked whether the expression pattern of *cut*, another well-known Notch target, was affected by N/Src. As expected, cut staining was upregulated in a broad swath when N$^{act}$ alone was driven by *vgGal4* (*Figure 7D*). Strikingly, this ectopic cut expression appeared to be completely absent in N$^{act}$+Src42A$^{CA}$ wing discs (*Figure 7B*). Furthermore, even endogenous cut disappeared in both N$^{act}$+Src42A$^{CA}$ (*Figure 7B*) and Src42A$^{CA}$ (*Figure 7C*) wing discs. Thus, cut is not upregulated like *E(spl)mγ* by N/Src, but rather suppressed like the other five *E(spl)* genes.

Since the major downstream effector of Notch is *Suppressor of Hairless (Su(H))* exerting its action by binding to Notch-dependent promoter sites, we asked if the NRE-GFP reporter, which is activated by Notch binding to Su(H) sites, could be activated by N/Src. We find that N$^{act}$+Src42A$^{CA}$ is still able to strongly activate the NRE-GFP reporter (*Figure 7F*).

These analyses demonstrate that N/Src synergy affects a diverse set of genes largely in a JNK dependent manner, including known Notch targets, which however seem to be differentially regulated.

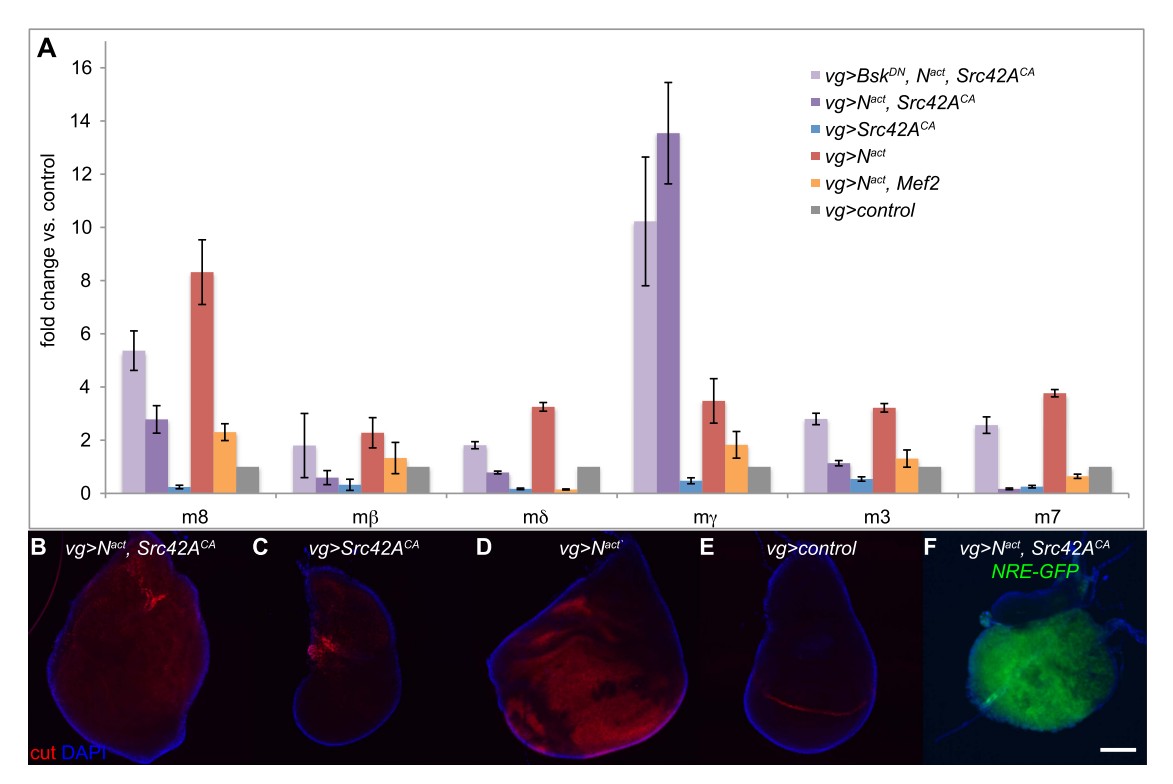

**Figure 7**. Notch targets are differentially affected by N/Src synergy. (**A**) qPCR assay for expression levels of *E(spl)* complex members in *vgGal4* wing discs expressing UAS constructs as indicated. (**B–E**) Immunostaining with anti-cut (red) in *vgGal4* wing discs. N$^{act}$ alone (**D**) induces cut expression, which is suppressed in N$^{act}$+Src42A$^{CA}$ discs (**B**). Note that both ectopic and endogenous cut appear to be suppressed. (**F**) NRE-GFP expression in wing discs expressing N$^{act}$+Src42A$^{CA}$ under *vgGal4*. Scale bar: 100 μm.

The following figure supplement is available for figure 7:

**Figure supplement 1**. E(spl)mγ reporter staining in N/Src wing discs.

## Discussion

The developmental outcome of a signaling pathway depends upon how the signal integrates with other pathways and factors within and between cells. Given that Notch defines an essential, ancient, and highly pleiotropic developmental pathway, it is plausible that the genetic circuitry that has evolved to influence the biological outcome of Notch signals is complex. The present analysis highlights the complexity and, significantly, the subtlety of Notch signal synergies. We focused our attention here on N/Src synergy partly because the proliferation phenotype we monitor depends, like our previously reported N/Mef2 synergy (*Pallavi et al., 2012*), on the activation of JNK signals. Importantly, however, we demonstrate that N/Src both accesses and interprets JNK signals in a manner distinct from that of N/Mef2. This sort of differential access to the same pathway to effect developmental outcomes may serve as a paradigm of the integration of a pleiotropic signal, such as Notch, with diverse, context-dependent factors. In turn, understanding the mechanisms of signal integration has profound implications for both normal development and disease.

### Notch interacts with a large, diverse cohort of genes to control proliferation

Notwithstanding the fact that the genetic circuitry surrounding Notch signals is very complex (*Guruharsha et al., 2012*), we were surprised to find such an abundance and diversity of genes that can cooperate with activated Notch signals to modulate cellular proliferation, a fact that has obvious potential consequences for oncogenesis as well as normal development.

It is important to note that our screen is not saturating because the Exelixis mutant collection disrupts only approximately 53% of the fly genome and has other inherent limitations, which have been described in detail elsewhere (*Thibault et al., 2004*; *Kankel et al., 2007*). In the current screen, we tested all lines (with the exception of insertions on the X chromosome, which would be hemizygous in male progeny) heterozygously, so weaker or recessive effects may not be observed. We also did not score combinations that were 100% lethal. Therefore, we do not expect to find every member of a complex or pathway or all redundant genes. In light of the above information, we presume that the true number of genes that can modify Notch-induced proliferation phenotypes is far greater (perhaps at least double) than the 360 we have identified in this work, indicating a surprising complexity and diversity of the potential genetic circuitry that can, in conjunction with Notch signals, affect proliferation.

Despite the diversity of the identified gene set, there are nevertheless some commonalities. Most notably, 31 genes are annotated as cell cycle genes, representing a more than twofold enrichment. In particular, genes involved in mitosis are highly represented. Given the phenotypic parameters of the screen, it is not surprising that cell cycle genes have been identified as modifiers. Interestingly, we identified *nanos* (*d06728*, which is predicted to be a GOF allele) as a suppressor of the large eye phenotype. This finding is corroborated by the observation that nanos can directly repress *Cyclin B (CycB)* in the germline, thereby preventing mitosis (*Kadyrova et al., 2007*). However, we also identified *c04775*, a predicted disruption of *Cyclin B3 (CycB3)*, as an enhancer; the evidence would suggest that loss of mitotic cyclins should, like *nanos* GOF, suppress rather than enhance hyperplasia. It is conceivable that this may point to a context-specific feedback regulation that, if indeed true, may have implications for oncogenesis. Such an interesting possibility, however, awaits experimental corroboration especially in view of the fact that additional analysis would be necessary to ensure that *c04775* is indeed a true loss-of-function mutation of *CycB3*.

We should also note that the screen did not always identify plausibly predicted genes. For example, *CycB*, which acts redundantly with *CycB3* in embryogenesis (*Lee and Orr-Weaver, 2003*), was not identified in our screen despite there being three predicted alleles in the Exelixis collection. On a similar note, we identified *Bub1 (c04512)* as a strong enhancer, but not any of the other key components of the spindle assembly checkpoint (*BubR1*, *Mad2*, and *Bub3*) (*Lara-Gonzalez et al., 2012*), even though they are represented in the Exelixis collection.

We cannot at present say if these observations imply some specificity of the Notch response for CycB3 and Bub1 or whether they simply reflect limitations of our screening system.

## The N/Src and N/Mef2 synergies converge on JNK but diverge in downstream output

Previous studies have demonstrated colocalization of Notch and c-Src proteins in pancreatic cancer cells, where Src is required for proteolytic activation of Notch (*Ma et al., 2012*), and of Notch and the T-cell-specific Src family member Lck in T cells (*Sade et al., 2004*), but a functional relationship between the two genes had not been demonstrated prior to our study. Given that the majority of N/Src phenotypes and changes in almost 70% of the transcriptional targets can be effectively reversed upon JNK inhibition it seems clear that the major target of the N/Src synergy is the JNK pathway. Our previously reported N/Mef2 synergy activates JNK via the TNF ligand *egr* (*Pallavi et al., 2012*), which is clearly not the case for N/Src, which actually causes suppression of both *egr* and its receptor. Additionally, N/Src causes a great deal of apoptosis, whereas N/Mef2 does not, and only N/Src is capable of differentially upregulating *E(spl)mγ*. Finally, although both combinations activate JAK/STAT ligands, the degree of upregulation is more than an order of magnitude greater for N/Src vs N/Mef2. These observations suggest that while JNK activation may be a common thread in Notch-related hyperplastic phenotypes, there may be multiple routes through which a single pathway (JNK) is activated, and this difference in access may subsequently contribute to differential downstream outputs (*Figure 8*).

It is worth noting that the Notch/JNK signaling axis may represent a primary pathway through which the cell can not only regulate proliferation but may also affect cell movement. Two groups have recently reported a role for actin polymerization upstream of JNK activation by Src (*Fernandez et al., 2014*; *Rudrapatna et al., 2014*); given that we observed increased actin in N/Mef2 tissues (*Pallavi et al., 2012*), we suggest that the same may be true for the N/Src axis.

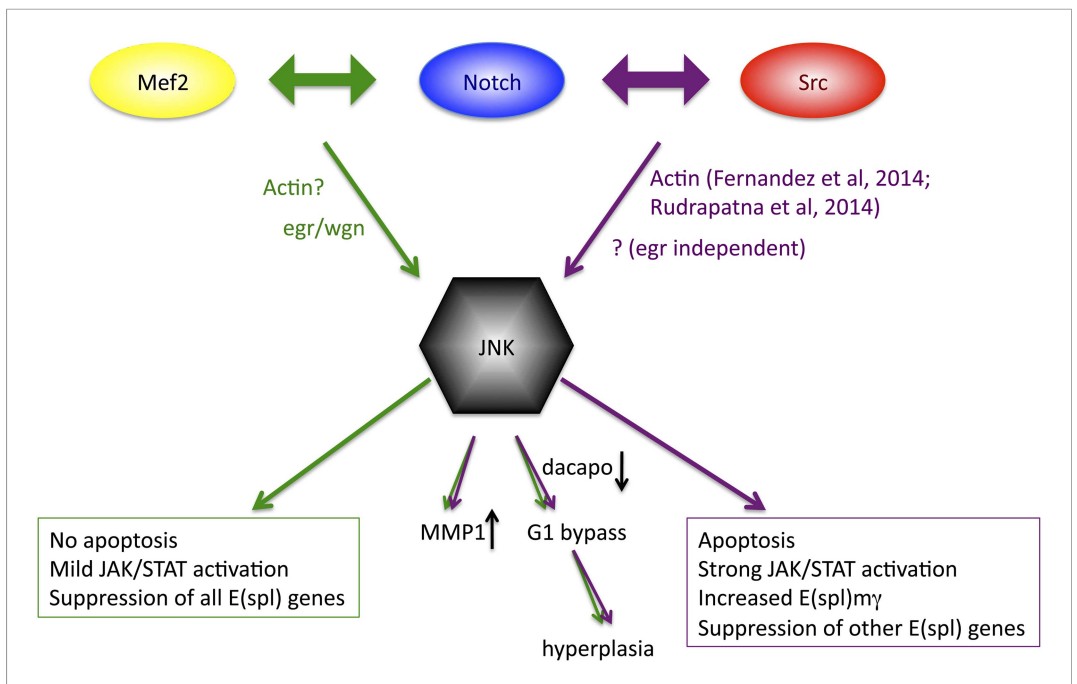

**Figure 8**. Model of convergence and divergence of the Notch/Mef2/JNK and Notch/Src/JNK signaling axes. N/Mef2 and N/Src synergies converge on JNK, through *eiger*-dependent and *eiger*-independent means respectively. Some downstream processes are common to both synergies, such as MMP1 activation and bypass of G1 phase of the cell cycle via *dap* downregulation. Other downstream outputs, such as apoptosis, level of JAK/STAT activation, and regulation of Notch target genes, diverge between N/Src and N/Mef2 synergy.

In addition to N/Src and N/Mef2, a third Notch synergy that acts through JNK has been reported: the loss of the epithelial polarity gene *scribble (scrib)* induces JNK activation leading to cell death; however, in the presence of active Notch, *scrib* mutant cells instead overgrow and become invasive (*Brumby and Richardson, 2003*). Whether N/scrib converges with either N/Src or N/Mef2 or defines a third JNK-dependent axis remains to be seen.

The vast diversity of signal outputs suggests that signal integration may occur not only at the level of Notch interactions but also at the level of JNK. Although the determinants of such specificity remain to be identified, we consider our identification of N/Src transcriptional targets to be a starting point for future studies.

## The differential regulation of Notch targets is a specific response to N/Src

Overexpression of Notch alone causes local expansion of the ZNC and distant hyperproliferation (*Johnston and Edgar, 1998*; *Herranz et al., 2008*). However, our data indicate that the combination of Notch and Src actually leads to loss of the ZNC and expansion of the hyperproliferative zone, suggesting that Src modulates the activity of Notch in this situation and leads to proliferation rather than differentiation (as is also implied by the loss of cut in the D–V boundary). Our observation of differential regulation of the *E(spl)* complex genes supports the notion that Src modulates the target specificity of Notch. The NRE-GFP reporter is strongly upregulated by N/Src (similar to Notch alone), suggesting that whatever leads to suppression of Notch targets in the presence of Src either does not affect Su(H) binding or requires the modulatory action of additional, differentially acting factors.

## Altered cell cycle profile as a result of N/Src synergy

Among the most striking N/Src phenotypes are the complete loss of the G1 phase of the cell cycle and an immense degree of hyperplasia. Our observation that the cell cycle regulator *dap* is downregulated by N/Src and N/Mef2 suggests a potential mechanism for the G1 bypass. The G1 phase is important

during normal development and homeostasis to allow newly-divided cells to increase in size and replenish protein and energy stores; additionally, regulatory checkpoints during G1 control cell fate choices such as the decision to exit the cell cycle and differentiate (*Lee and Orr-Weaver, 2003*; *Hindley and Philpott, 2013*). *Dap* is a member of the p21/p27 family of Cdk inhibitors, and is required during *Drosophila* embryogenesis for proper timing of cell cycle exit at G1 (*de Nooij et al., 1996*; *Lane et al., 1996*).

Early *Drosophila* embryos undergo their first 16 cell divisions without G1 or G2, resulting in rapid cell division without a corresponding increase in compartment size (*Lee and Orr-Weaver, 2003*). In contrast, we observe hyperplasia of N/Src wing discs, indicating that compartment growth must occur even in the absence of a detectable G1 phase. In the case of Ras or Myc overexpression, the other phases of the cell cycle are prolonged to compensate for decreased time in G1, and the overall growth rate also increases (*Johnston et al., 1999*; *Prober and Edgar, 2000, 2002*). We do observe an increase in the percentage of N/Src cells in S phase, suggesting that the lack of G1 is at least partially compensated for by a longer S.

The N/Src cell cycle profile, with absent G1 and a high proportion of cells in S phase, is reminiscent of that of mammalian embryonic stem cells (ESCs), which have an abbreviated G1 and longer S; recent studies have suggested that this altered cycle is necessary to maintain pluripotency, although the mechanisms are still unclear (*Orford and Scadden, 2008*; *Hindley and Philpott, 2013*). Like ESCs, N/Src cells remain in an actively proliferating state rather than undergoing G1 arrest followed by differentiation. Therefore, this raises the very interesting possibility that N/Src synergy may interfere with cell fate determination, conferring pluripotency. This could have important implications for tumorigenicity as well as for the maintenance of a differentiated state. A recent report found that Src42A and Src64B in *Drosophila* intestinal stem cells (ISCs) could stimulate the expansion of a transit-amplifying population of Notch-positive cells (*Kohlmaier et al., 2014*). In light of our findings, we suggest that Src may actually interact with Notch in this cell population to induce proliferation. It will therefore be informative to compare our list of N/Src transcriptional targets with the genes induced by Src in ISCs or other pluripotent cell types in both mammals and flies.

## Implications for cancer biology

Several previous studies have shown that Src acts through JNK to induce invasive phenotypes, including activation of MMP1 (*Ma et al., 2014*; *Rudrapatna et al., 2014*). However, invasiveness is only one part of the oncogenic equation. Once they have migrated into a tissue, metastatic cells also need to grow and proliferate, preferably in an unchecked fashion. Our work suggests that the addition of Notch promotes hyperplastic growth while still retaining and perhaps even enhancing Src-driven invasive behavior. Strikingly, this occurs even in the presence of significant amounts of apoptosis. Should an additional mutation or gene activation occur that suppresses this cell death, the N/Src cells may become even more malignant. A clue might be found in the comparison with N/Mef2, where tissues overgrow but very little cell death is observed.

Other studies have reported that loss of the Src antagonist Csk in the wing disc can induce activation of JNK and JAK/STAT signaling as well as morphological disorganization (*Read et al., 2004*; *Vidal et al., 2006*; *Langton et al., 2007*; *Vidal et al., 2007*). We hypothesize that these Csk phenotypes may result from activation of endogenous Src, which then interacts with endogenous Notch. The Src family members *Src*, *Fyn*, and *Yes* in mammals, as well as *Src42A* and *Src64B* in *Drosophila*, are widely, even ubiquitously expressed, and kept in check partly by the similarly widespread expression of *Csk* (*Takahashi et al., 1996*; *Tateno et al., 2000*; *Takahashi et al., 2005*; *Wheeler et al., 2009*). Notch is likewise expressed in a large number of tissues (*Artavanis-Tsakonas et al., 1999*; *Artavanis-Tsakonas and Muskavitch, 2010*).

Interestingly, although the activity and levels of both Src and Notch are often increased in cancers, activating mutations in the genes themselves are rare (*Ishizawar and Parsons, 2004*; *Ranganathan et al., 2011*; *Louvi and Artavanis-Tsakonas, 2012*). One explanation suggested by our data is that synergy between even relatively low levels of active Notch and Src can activate JNK and cause exponentially increased levels of hyperplastic growth and stimulation of oncogenic events. Additionally, an inactivating mutation in Csk could have devastating consequences in both development and cancer by triggering a Notch/Src synergistic response.

Among the human orthologs of our N/Src targets, genes involved in metabolism and stress response are both overrepresented. Thus these cells may have an increased metabolic rate (which may

also explain the observed overgrowth in the absence of G1) and heightened protection from stress, both of which could contribute to a favorable environment for oncogenesis. In particular, this may explain how N/Src cells survive and indeed hyperproliferate despite the strong pro-apoptotic JNK signal.

An earlier study revealed that activated Notch could act as a tumor suppressor in v-Src-transformed quail neuroepithelial cells, causing a reversion of the transformed phenotype along with suppression of JNK signaling (*Mateos et al., 2007*). This report may seem on the surface to contradict our findings. However, Notch can act as either an oncogene and a tumor suppressor depending on context (*Lobry et al., 2014*). Here, too, context, including other genes that interact with the N/Src axis, is likely to be essential; further analysis of the hits from our genetic screen may shed light on this issue as well as on the larger question of how Notch switches between oncogenic and tumor suppressive behavior.

Our observation that the N/Mef2 and N/Src synergies both activate the same pathway (JNK) but display differences in downstream phenotype suggests that it may be possible to identify targeted, refined signatures for different types of Notch, Src, and JNK-related cancers.

While single gene mutations can occasionally trigger an oncogenic state, it is more often the context-dependent interplay between genes that causes or modulates cancerous growth. Understanding the mechanistic consequences of cross-talking gene activities is essential if we are to unveil the molecular basis of oncogenic events and develop rational therapeutic interventions.

## Materials and methods

### Fly strains

Crosses were carried out at 25°C under standard conditions unless otherwise noted. Exelixis lines used in this work can be obtained at https://drosophila.med.harvard.edu/ (*Artavanis-Tsakonas, 2004*; *Parks et al., 2004*; *Thibault et al., 2004*). Other fly lines used were: UAS-N$^{act}$ (*Go et al., 1998*), UAS-Src42A$^{CA}$ (*Tateno et al., 2000*), UAS-Src64B (*Nicolai et al., 2003*), puc$^{E69}$ (*Martin-Blanco et al., 1998*), UAS-Bsk$^{DN}$ (*Adachi-Yamada et al., 1999*), upd-LacZ (gift from N Perrimon), dap$^{k07309}$ (*Mitchell et al., 2010*), 10XStatGFP (*Bach et al., 2007*), N$^{55e11}$ (*Rulifson and Blair, 1995*), E(spl)mγKX-LacZ (*Nellesen et al., 1999*), NRE-GFP (*Saj et al., 2010*), UAS-Mef2 (*Bour et al., 1995*), and UAS-Notch$^{RNAi}$ (*Hori et al., 2004*). Gal4 lines used were E1Gal4 (gift from G Rubin [*Pallavi et al., 2012*]), vgGal4, MS1096Gal4 and dppGal4 (all available from the Bloomington *Drosophila* Stock Center, Bloomington, IN).

### Antibodies and staining

Immunostaining and EdU staining were performed as described in our previous work (*Pallavi et al., 2012*). The following primary antibodies were used: 3A6B4 anti-MMP1 (1:100; Developmental Studies Hybridoma Bank (DSHB), Iowa City, IA), anti-cleaved caspase 3 (1:300; Cell Signaling Technology, Danvers, MA), D5.1 anti-GFP (1:300; Cell Signaling Technology), anti-β-gal (1:1000-1:2000, MP Biomedicals, Santa Ana, CA), anti-Src-pY418 (1:1000, Abcam, Cambridge, MA) and 2B10 anti-cut (1:10, DSHB). AlexaFluor conjugated secondary antibodies (Life Technologies, Carlsbad, CA) were used at 1:1000. Fluorescence microscopy (with the exception of the EdU incorporation assay) was performed on a Zeiss Axioplan microscope with a 10× objective and images were minimally processed using Adobe Photoshop CS5. The discs for the EdU incorporation assay (*Figure 5*) were imaged using a Nikon TE2000 with C1 Point Scanning Confocal at the Nikon Imaging Center at Harvard Medical School.

### Genetic screen

UAS-N$^{act}$/CyO-tub-Gal80; E1Gal4 fly stocks were generated and virgins were crossed to males from each line in the Exelixis mutant collection (*Artavanis-Tsakonas, 2004*; *Thibault et al., 2004*; *Pallavi et al., 2012*) to screen for enhancers and suppressors of the Notch-induced large eye phenotype. All positive hits were rescreened a second time to eliminate false positives; furthermore, they were crossed to E1Gal4 alone to eliminate Notch-independent effects. We identified 332 Exelixis lines that either enhanced or suppressed the large eye phenotype. The determination of genes predicted to be affected by each Exelixis line was performed as previously described (*Sen et al., 2013*). In some cases, more than one gene may be affected either due to multiple insertions or to insertion of an element in

or near overlapping or neighboring genes. Mapping of *Drosophila* genes to human orthologs was performed using tables generated by Mark Gerstein's group at Yale University, where *Drosophila*-human ortholog pairs were identified based on three sources—InParanoid, OrthoMCL and TreeFarm (http://info.gersteinlab.org/Ortholog_Resources). Here, we provide only the top human ortholog (identified by the most number of sources) for each *Drosophila* gene. GO term enrichment analysis was performed using DAVID (http://david.abcc.ncifcrf.gov/) with the total Exelixis gene list as background; statistical significance was determined using the Benjamini–Hochberg correction. The interaction map shown in *Figure 1C* was generated using GeneMania (www.genemania.org), using all available default datasets for genetic interactions, physical interactions, predicted interactions, and shared protein domains, and a limit of 20 related genes.

## DNA content analysis

Wing discs from wandering third instar larvae were collected in 1× PBS and dissociated in 9× Trypsin-EDTA (Life Technologies) with 0.5 µg/ml Hoechst 33,342 and 1× PBS for approximately 4 hr at room temperature with gentle agitation (*de la Cruz and Edgar, 2008*). FACS was performed on dissociated cells using a BD FACSAria II SORP UV, and data was analyzed using FlowJo and ModFit software.

## qPCR

Total RNA for qPCR was extracted from wing discs from wandering third instar larvae. Discs were isolated in PBS and RNA was extracted using TRIzol reagent (Life Technologies); gDNA was removed and RNA was cleaned up using the RNEasy plus micro kit (Qiagen, Valencia, CA). cDNA was generated using the High Capacity RNA to cDNA kit (Life Technologies). All qPCR reactions were performed in technical triplicate using Taqman assays (Life Technologies) on a Life Technologies 7900HT machine. Unless otherwise noted, at least three biological replicates were performed for each genotype.

## RNA-seq

*VgGal4* females were crossed to males from the UAS lines of interest. Total RNA for RNA-seq was isolated and purified as for qPCR from at least 60 wing discs per sample. All samples were run in biological triplicate. Ribosomal RNA depletion was performed using the Ribo-zero rRNA removal kit (Epicentre, Madison, WI). Library preparation using the PrepX SPIA RNA-seq kit (IntegenX, Pleasanton, CA), library quality control, and sequencing on an Illumina HiSeq2000 machine was performed by the Biopolymers Facility at Harvard Medical School (http://genome.med.harvard.edu). We ran 100 cycle paired-end reads for each sample. Sequencing data was analyzed using Galaxy (http://usegalaxy.org/) (*Giardine et al., 2005*; *Goecks et al., 2010*; *Blankenberg and Hillman-Jackson, 2014*). Specifically, we used TopHat to align reads to the *Drosophila melanogaster* reference genome (BDGP R5/dm3) and CuffDiff with replicate analysis using a false discovery rate of 0.05 and no additional parameters to compare sample pairs (*Garber et al., 2011*; *Trapnell et al., 2012*). Genes were defined as synergistically up- or down- regulated by N/Src only if the adjusted p-value was less than 0.05 for all of the following pairs: N/Src vs N, N/Src vs Src, and N/Src vs WT control. Synergistic targets were further defined as rescued by Bsk[DN] if the adjusted p-value for Bsk[DN]/N/Src vs N/Src was less than 0.05. Mapping of *Drosophila* genes to human orthologs was performed as for screen hits above.

## Acknowledgements

We thank Sue Celniker and Mike Duff for advice on RNA-seq, Bob Obar for help in determination of human orthologs of *Drosophila* genes, Norbert Perrimon, Gerald Rubin, James Posakony, and the Bloomington *Drosophila* Stock Center (NIH P40OD018537) for fly stocks, the Developmental Studies Hybridoma Bank for antibodies, and Wade Harper, Joan Brugge, and Lucio Miele for critical reading of the manuscript. DMH was supported by the American Cancer Society New England Division-Virginia Cochary Award for Excellence in Breast Cancer Research Postdoctoral Fellowship (PF-10-230-01-DDC). This research was partly funded by NIH grant CA 098402 to S A-T.

## Additional information

### Funding

| Funder | Grant reference | Author |
| --- | --- | --- |
| American Cancer Society | Research Postdoctoral Fellowship (PF-10-230-01-DDC) | Diana M Ho |
| National Institutes of Health | CA 098402 | Spyros Artavanis-Tsakonas |

The funders had no role in study design, data collection and interpretation, or the decision to submit the work for publication.

### Author contributions

DMH, Conceived, designed, performed, and analyzed all experiments; wrote manuscript., Conception and design, Acquisition of data, Analysis and interpretation of data, Drafting or revising the article; SKP, Conceived, designed and performed experiments for the genetic screen., Conception and design, Acquisition of data; SA-T, Conceived and designed experiments; revised manuscript., Conception and design, Drafting or revising the article

## Additional files

### Supplementary files

• Supplementary file 1. List of Exelixis mutant lines that enhance or suppress the *E1Gal4; UAS-N$^{act}$*-induced large eye phenotype. Lines in bold appear more than once in this table as they may affect more than one target gene. 26 of these lines were retested using qPCR for *puc* and *MMP1* in *vgGal4; UAS-N$^{act}$* wing discs. These genes were determined to be upregulated if the fold change by qPCR vs isogenic wild-type controls was at least 1.5. *MMP1* upregulation was further verified by immunostaining.

• Supplementary file 2. GO term enrichment analysis for Notch interactors. Enrichment analysis was performed with DAVID (http://david.abcc.ncifcrf.gov), using the total Exelixis gene list as background; corrected p-values are shown using both the Benjamini–Hochberg and Bonferroni corrections.

• Supplementary file 3. List of genes synergistically regulated by N$^{act}$ and Src42A$^{CA}$ as determined by RNA-seq analysis (P-adj < 0.05 for NS vs N, S, and WT). Genotypes: NS: *vgGal4/UAS-N$^{act}$; UAS-Src42A$^{CA}$/+*, S: *vgGal4/+; UAS-Src42A$^{CA}$/+*, N: *vgGal4/UASN$^{act}$*, WT: *vgGal4/+*, BNS: *UAS-Bsk$^{DN}$/+, vgGal4/UAS-N$^{act}$, UAS-Src42A$^{CA}$/+*.

• Supplementary file 4. RNA-seq data for known N/Src targets. Genotypes: NS: *vgGal4/UAS-N$^{act}$; UAS-Src42A$^{CA}$/+*, S: *vgGal4/+; UAS-Src42A$^{CA}$/+*, N: *vgGal4/UASN$^{act}$*, WT: *vgGal4/+*, BNS: *UAS-Bsk$^{DN}$/+, vgGal4/UAS-N$^{act}$, UAS-Src42A$^{CA}$/+*.

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
