## [Decision Letter]

Thank you for sending your work entitled “The Notch-mediated hyperplasia circuitry in *Drosophila* reveals a Src-JNK signaling axis” for consideration at *eLife*. Your article has been favorably evaluated by K VijayRaghavan (Senior editor and Reviewing editor) and two reviewers.

The Reviewing editor and the reviewers discussed their comments before we reached this decision, and the Reviewing editor has assembled the following comments to help you prepare a revised submission.

It has become increasing obvious that the driving force for many cancers involves combinations of signalling pathways. Identifying fundamental synergies between pathways is therefore of major significance. In this manuscript, the authors investigate the consequences from combined activity of Notch and Src. The manuscript reports the results of a genome-wide genetic screen to identify modifiers of Notch-mediated hyperplasia in *Drosophila*. From the list of modifiers, the authors focus their attention on Src and in the involvement of JNK on the N/Src synergy. A valuable contribution is the screen itself whose results will be welcomed by the research community. Therefore, giving more prominence to the screen and its results, compared to the Src/N synergy (which is important too, see below) could make the paper's general value more apparent.

The experiments reveal that the phenotypes from combining ectopic Notch (expressing N^act^) and from ectopic Src (expressing various forms) are considerably more severe than either alone. Notably the tissues exhibit exacerbated hyperplasia. Additional experiments demonstrate that there are changes in cell cycle regulation, enhanced apoptosis and altered expression of a number of different types of downstream genes including *MMP1*, *decapo*, JAK-STAT ligands (*upds*). The authors also perform RNA-seq to characterize fully the transcriptome of the different genotypes. These are therefore substantial and valuable studies that are analysed but also provide data to the community for further mechanistic studies.

There are some intriguing aspects of the data from the screen that the authors could elaborate as discussed below.

An interesting feature of the manuscript is the demonstration that synergy between Notch and Src42A in the eye and wing discs causes hyperplasia and activates JNK, and that halved Notch activity can rescue the phenotype of over-expressed Src: the striking result is the rescue of Src64B phenotype in N/+ heterozygotes. These data suggest that Src64B requires Notch for many actions, potentially a profound observation. One concern about this, and some other experiments, is that they rely on *vg-Gal4* driver. As *vg* is regulated by Notch activity, there is a possibility that the suppression in N/+ and the augmented phenotypes with N^act^ are due to changes in Gal4 expression, leading to less or more Src being expressed respectively. At least some of the experiments should be repeated with a Notch-independent driver (e.g. engrailed-Gal4) to rule out this possibility. Or alternative ways of addressing this concern need to be elaborated. This important issue bears reiterating: Given the likelihood that *vg-GAL4* is influenced by Notch activity, at least some of the core findings, including this one, should be reproduced with another strategy.

Several of the results from the screen are intriguing and merit discussion. For example, the authors could elaborate on their finding of *nanos (d06728)* as a suppressor of somatic hyperplasia. A second interesting aspect of the results of the screening is the involvement of *CycB3 (d04775)*. It is remarkable that loss of function of a mitotic cyclin can enhance Notch-mediated hyperplasia and it is striking that none of the other cyclins appears as a modifier. Something similar applies to *Bub1 (c04512)*, which shows up as a strong enhancer while none of the other gatekeepers of the metaphase to anaphase transition (*Mad3/BubR1, Mad2,* or *Bub3*) do so.

Previous studies investigating Src64 reported some similar phenotypes (MMP1, apoptosis, JNK activity; Cell Death Dis.;4:e864; Oncogene.33(16):2027-39). It would be helpful to place the current results in the context of the previous studies of ectopic Src, which also clearly demonstrated that JNK was required. It is therefore not unexpected that bsk^DN^ blocks many of these phenotypes (as shown previously). What is unclear is how N^act^ augments these Src phenotypes. This remains a major gap in the current work that should be discussed in a revised version.

The RNA-seq experiment does not appear to support other qPCR/ immunofluorescence data in the manuscript: *MMP1*, *puc*, *dap*, *upd* are not included in the genes synergistically regulated. Do the authors have an explanation? As it stands these data undermine their conclusions. Use of additional methods to substantiate core targets (*dap*, *upd*, *E(spl)mγ*) would strengthen their conclusions (e.g. in situ hybridisation or reporter genes ) and would give useful insight into whether the effects are autonomous.

The data showing effects on the Zone of non-proliferating cells (ZNC) in Figure 4 are not sufficiently clear. The disc appears to totally lack a ZNC. Is this because of a broad non-autonomous effect or is the disc younger?

Additional controls would strengthen the conclusions. Most studies of this type control for the numbers of UAS present (using *UAS-GFP* or *UAS-lacZ* etc.) and also mark the region of ectopic expression (e.g. Oncogene. 33(16):2027-39). Although it is unlikely that UAS titration has confounded the results here (in most experiments there is an increased phenotype) there are some genotypes where this could be a factor. In addition, for Figure 7, controls without the balancer chromosomes are needed.

Mateos et al. (Oncogene, 2007; doi:10.1038/sj.onc.1210124) have previously shown in avian neural cells that intracellular Notch suppresses v-Src-induced transformation and documented that such a suppression correlates with changes in JNK-dependent signalling. Mateos et al. should be cited and discussed. A previous manuscript from the same group on the Notch/Mef2 synergy (Pallavi1, et al. 2012, EMBO J) partially overlaps with the submitted manuscript. This matter should be addressed in revision.

---

## [Author Response]

*It has become increasing obvious that the driving force for many cancers involves combinations of signalling pathways. Identifying fundamental synergies between pathways is therefore of major significance. In this manuscript, the authors investigate the consequences from combined activity of Notch and Src. The manuscript reports the results of a genome-wide genetic screen to identify modifiers of Notch-mediated hyperplasia in* Drosophila*. From the list of modifiers, the authors focus their attention on Src and in the involvement of JNK on the N/Src synergy. A valuable contribution is the screen itself whose results will be welcomed by the research community. Therefore, giving more prominence to the screen and its results, compared to the Src/N synergy (which is important too, see below) could make the paper's general value more apparent*.

We fully agree with you that the results of the screen will be valuable to the community and welcome your comment. As such, we have added a deeper discussion of these results (both in the Results and Discussion sections). We have added a panel to Figure 1 (Figure 1) that shows preexisting known and predicted interactions between all cell cycle genes from the screen; this reveals that we have identified novel links between these genes and Notch. We’ve also added significantly to the Discussion regarding the nature of the collection and our evaluation of the strengths/weaknesses thereof. Specifically, we think it’s crucial to convey the fact that the screen was not saturating; therefore we do not expect to recover all of the possible modifiers of Notch. Finally, we’ve discussed some interesting specific genes/pathways from the screen, including the *nanos/cyclinB3* connection mentioned below.

*The experiments reveal that the phenotypes from combining ectopic Notch (expressing N*^*act*^*) and from ectopic Src (expressing various forms) are considerably more severe than either alone. Notably the tissues exhibit exacerbated hyperplasia. Additional experiments demonstrate that there are changes in cell cycle regulation, enhanced apoptosis and altered expression of a number of different types of downstream genes including* MMP1*,* decapo*, JAK-STAT ligands (*upds*). The authors also perform RNA-seq to characterize fully the transcriptome of the different genotypes. These are therefore substantial and valuable studies that are analysed but also provide data to the community for further mechanistic studies*.

*There are some intriguing aspects of the data from the screen that the authors could elaborate as discussed below*.

*An interesting feature of the manuscript is the demonstration that synergy between Notch and Src42A in the eye and wing discs causes hyperplasia and activates JNK, and that halved Notch activity can rescue the phenotype of over-expressed Src: the striking result is the rescue of Src64B phenotype in N/+ heterozygotes. These data suggest that Src64B requires Notch for many actions, potentially a profound observation. One concern about this, and some other experiments, is that they rely on* vg-Gal4 *driver. As* vg *is regulated by Notch activity, there is a possibility that the suppression in N/+ and the augmented phenotypes with N*^*act*^
*are due to changes in Gal4 expression, leading to less or more Src being expressed respectively. At least some of the experiments should be repeated with a Notch-independent driver (e.g. engrailed-Gal4) to rule out this possibility. Or alternative ways of addressing this concern need to be elaborated. This important issue bears reiterating: Given the likelihood that* vg-GAL4 *is influenced by Notch activity, at least some of the core findings, including this one, should be reproduced with another strategy*.

For the overexpression experiments, we have used *dppGal4*, which is independent of Notch, to validate our *vgGal4* results. We find that *dppGal4*-driven N^act^/Src also causes upregulation of MMP1 (Figure 4), increased apoptosis (Figure 4), and cell cycle perturbation (Figure 5 and Figure 5—figure supplement 1).

Regarding the rescue experiment with *N*^*55e11*^*/+* and *UAS-N*^*RNAi*^, we have attempted to perform this experiment in a vg-independent manner using several approaches but this has failed. Specifically, we have tried to use both *dppGal4* and *C96Gal4* wing drivers, as well as both the *Csk*^*j1d8/j1d8*^ mutant (which inactivates Csk, an inhibitor of Src) and the *Csk*^*j1d8*^*/+,puc*^*E69*^*/+* trans-heterozygous mutant, which were both reported by Langton et al. (Dev Cell, 2007) to cause Src-like phenotypes in the wing disc. Unfortunately, all of the above approaches failed due to excessive lethality and we were thus unable to replicate the rescue experiment using a vg-independent method. We have therefore decided to move this result into the supplemental data (it is now Figure 3—figure supplement 2) and explicitly state in the text the caveat that effects of Notch reduction on *vgGal4* may play a role. We also avoid drawing any conclusions in the Discussion based on this finding.

*Several of the results from the screen are intriguing and merit discussion. For example, the authors could elaborate on their finding of* nanos *(*d06728*) as a suppressor of somatic hyperplasia. A second interesting aspect of the results of the screening is the involvement of* CycB3 (d04775)*. It is remarkable that loss of function of a mitotic cyclin can enhance Notch-mediated hyperplasia and it is striking that none of the other cyclins appears as a modifier. Something similar applies to* Bub1 (c04512)*, which shows up as a strong enhancer while none of the other gatekeepers of the metaphase to anaphase transition (*Mad3/BubR1, Mad2, *or* Bub3*) do so*.

As mentioned above, we have added substantial discussion of the results from the genetic screen, including mention of *cyclin B3* and *nanos*. With regard to only single members of a pathway or process showing up in the screen, it is important not to read too much into such things as they may be caused by the fact that the screen is not saturating (this and other limitations of the screen are now discussed). However, we note that *BubR1, Mad2*, and *Bub3* are indeed represented in the Exelixis collection and were not identified as modifiers in our screen; we now discuss this as a potentially interesting avenue for further study.

*Previous studies investigating Src64 reported some similar phenotypes (MMP1, apoptosis, JNK activity; Cell Death Dis.;4:e864; Oncogene.33(16):2027-39). It would be helpful to place the current results in the context of the previous studies of ectopic Src, which also clearly demonstrated that JNK was required. It is therefore not unexpected that bsk*^*DN*^
*blocks many of these phenotypes (as shown previously). What is unclear is how N*^*act*^
*augments these Src phenotypes. This remains a major gap in the current work that should be discussed in a revised version*.

This was indeed an omission. We’ve now included these studies in our Discussion. While they do indeed show that Src activates JNK/MMP1/apoptosis to promote invasive behavior, they do not document hyperplastic growth. We believe that the addition of Notch is important to contribute the equally important overgrowth aspect of oncogenesis. It is striking that even in the presence of such massive apoptosis as induced by JNK, N/Src tissues can still overgrow.

*The RNA-seq experiment does not appear to support other qPCR/ immunofluorescence data in the manuscript:* MMP1*,* puc*,* dap*,* upd *are not included in the genes synergistically regulated. Do the authors have an explanation? As it stands these data undermine their conclusions*.

We have included a new table ([Supplementary-material SD4-data]) containing the RNA-seq data for these genes. For *MMP1* and the *upd* genes, the discrepancy can be largely explained by the observation that Src alone can induce these genes to a lesser degree. This seems to sometimes be enough for the algorithm/false positive correction to deem the N/Src vs. Src condition statistically insignificant. For *dap*, we see a significant reduction of the Notch-induced *dap* in the presence of Src; however, we do not see the reduction below WT that we see with qPCR. This discrepancy remains unexplained at the moment; we make note of this fact in the text. With regards to *puc*, we believe that the RNA-seq gave a false positive reading for the WT condition that threw off the subsequent calculations. This is explicitly now mentioned in the text.

*Use of additional methods to substantiate core targets (*dap, upd, E(spl)mγ*) would strengthen their conclusions (e.g. in situ hybridisation or reporter genes ) and would give useful insight into whether the effects are autonomous*.

We have performed experiments using the *LacZ* reporter lines for *upd*, *dap*, and *E(spl)mγ* as you suggested. The new results are shown in Figures 5 and 6, and Figure 7—figure supplement 1 respectively and support our conclusions.

*The data showing effects on the Zone of non-proliferating cells (ZNC) in*
Figure 4
*are not sufficiently clear. The disc appears to totally lack a ZNC. Is this because of a broad non-autonomous effect or is the disc younger*?

We’ve replaced these images with confocal images that are clearer and better stage-matched. You are correct in that there is some non-autonomous effect on proliferation induced by both N alone and N/Src that causes distortion (specifically, expansion of the dorsal part of the wing pouch), and we’ve now pointed this out in the text as well.

*Additional controls would strengthen the conclusions. Most studies of this type control for the numbers of UAS present (using* UAS-GFP *or* UAS-lacZ *etc.) and also mark the region of ectopic expression (e.g. Oncogene. 33(16):2027-39). Although it is unlikely that UAS titration has confounded the results here (in most experiments there is an increased phenotype) there are some genotypes where this could be a factor*.

We have performed a control experiment where we have compared discs containing one copy each of *UAS-GFP, UAS-N*^*act*^, *and UAS-Src42A*^*CA*^ (3 UAS transgenes total) to discs containing only *UAS-N*^*act*^ and *UAS-Src42A*^*CA*^ (2 UAS transgenes total). We find that there is no difference in phenotype between the two, indicating that the rescue we see when we add *UAS-Bsk*^*DN*^ to *UAS-N*^*act*^ and *UAS-Src42A*^*CA*^ is not a result of Gal4/UAS titration. This result is shown in Figure 3—figure supplement 1.

With regards to marking the region of ectopic expression, in the experiments using the *dppGal4* driver (Figure 4, as well as the existing ZNC experiment in Figure 5), we have included *UAS-GFP* to mark the expression domain.

*In addition, for*
Figure 7*, controls without the balancer chromosomes are needed*.

For this experiment (now Figure 3—figure supplement 2), a control without the balancer chromosome has been added in panel D.

*Mateos et al. (**Oncogene, 2007;*
*doi:10.1038/sj.onc.1210124**) have previously shown in avian neural cells that intracellular Notch suppresses v-Src-induced transformation and documented that such a suppression correlates with changes in JNK-dependent signalling. Mateos et al. should be cited and discussed*.

Indeed! We have cited and discussed this report as you suggested. We note that Notch can act as both an oncogene and a tumor suppressor depending on context, and this may explain the discrepancy between this study and the present one.

*A previous manuscript from the same group on the Notch/Mef2 synergy (Pallavi1, et al. 2012, EMBO J) partially overlaps with the submitted manuscript. This matter should be addressed in revision*.

Our previous paper has been provided to *eLife* as a related manuscript. The overlapping data (showing N/Mef2 activity on MMP1, apoptosis, and JNK) is clearly described in the text as replicating our previous work; we believe that it is important to show it in the current work for comparison/control purposes.